**DOI: 10.1038/ncomms13724**　　**OPEN**

# Therapeutic microparticles functionalized with biomimetic cardiac stem cell membranes and secretome

Junnan Tang[1,2,3,4,*], Deliang Shen[1,4,*], Thomas George Caranasos[5,*], Zegen Wang[6], Adam C. Vandergriff[2,3], Tyler A. Allen[2,3], Michael Taylor Hensley[2,3], Phuong-Uyen Dinh[2,3], Jhon Cores[2,3], Tao-Sheng Li[7], Jinying Zhang[1,4], Quancheng Kan[8] & Ke Cheng[2,3,6,9]

Stem cell therapy represents a promising strategy in regenerative medicine. However, cells need to be carefully preserved and processed before usage. In addition, cell transplantation carries immunogenicity and/or tumourigenicity risks. Mounting lines of evidence indicate that stem cells exert their beneficial effects mainly through secretion (of regenerative factors) and membrane-based cell–cell interaction with the injured cells. Here, we fabricate a synthetic cell-mimicking microparticle (CMMP) that recapitulates stem cell functions in tissue repair. CMMPs carry similar secreted proteins and membranes as genuine cardiac stem cells do. In a mouse model of myocardial infarction, injection of CMMPs leads to the preservation of viable myocardium and augmentation of cardiac functions similar to cardiac stem cell therapy. CMMPs (derived from human cells) do not stimulate T-cell infiltration in immuno-competent mice. In conclusion, CMMPs act as 'synthetic stem cells' which mimic the paracrine and biointerfacing activities of natural stem cells in therapeutic cardiac regeneration.

[1] Department of Cardiology, The First Affiliated Hospital of Zhengzhou University, Zhengzhou, Henan 450052, China. [2] Department of Molecular Biomedical Sciences and Comparative Medicine Institute, North Carolina State University, Raleigh, North Carolina 27607, USA. [3] Department of Biomedical Engineering, University of North Carolina at Chapel Hill & North Carolina State University, Chapel Hill and Raleigh, North Carolina 27599 and 27607, USA. [4] Institute of Clinical Medicine, The First Affiliated Hospital of Zhengzhou University, Zhengzhou, Henan 450052, China. [5] Division of Cardiothoracic Surgery, University of North Carolina at Chapel Hill, Chapel Hill, North Carolina 27599, USA. [6] The Cyrus Tang Hematology Center, Soochow University, Suzhou, Jiangsu 215123, China. [7] Department of Stem Cell Biology, Atomic Bomb Disease Institute, Nagasaki University, 1-12-4 Sakamoto, Nagasaki 852-8523, Japan. [8] Department of Pharmacy, The First Affiliated Hospital of Zhengzhou University, Zhengzhou, Henan 450052, China. [9] Division of Pharmacoengineering and Molecular Pharmaceutics, Eshelman School of Pharmacy, University of North Carolina at Chapel Hill, Chapel Hill, North Carolina 27599, USA. * These authors contributed equally to this work. Correspondence and requests for materials should be addressed to J.Z. (email: jyzhang@zzu.edu.cn) or to Q.K. (email: kanqc@zzu.edu.cn) or to K.C. (email: ke_cheng@ncsu.edu).

Multiple types of adult stem cells, such as mesenchymal stem cells, cardiac stem cells (CSCs), and endothelial progenitor cells have entered clinical investigations worldwide[1–6]. Differentiation of injected cells into the host tissues has been reported. However, these sporadic events could not explain the therapeutic benefits seen in animal models and human trials[7,8]. Later on, the field realized that one important mode of therapeutic action is the secretion of paracrine factors by injected stem cells that act like 'mini-drug pumps' to promote endogenous repair[9,10]. Moreover, stem cell membranes are not null in the repair process: contact with the injected stem cells triggers intracellular protective/regenerative pathways in the host cells[11,12]. On the basis of these two aspects, we proposed a 'core-shell' design of a therapeutic microparticle (MP) which mimicked

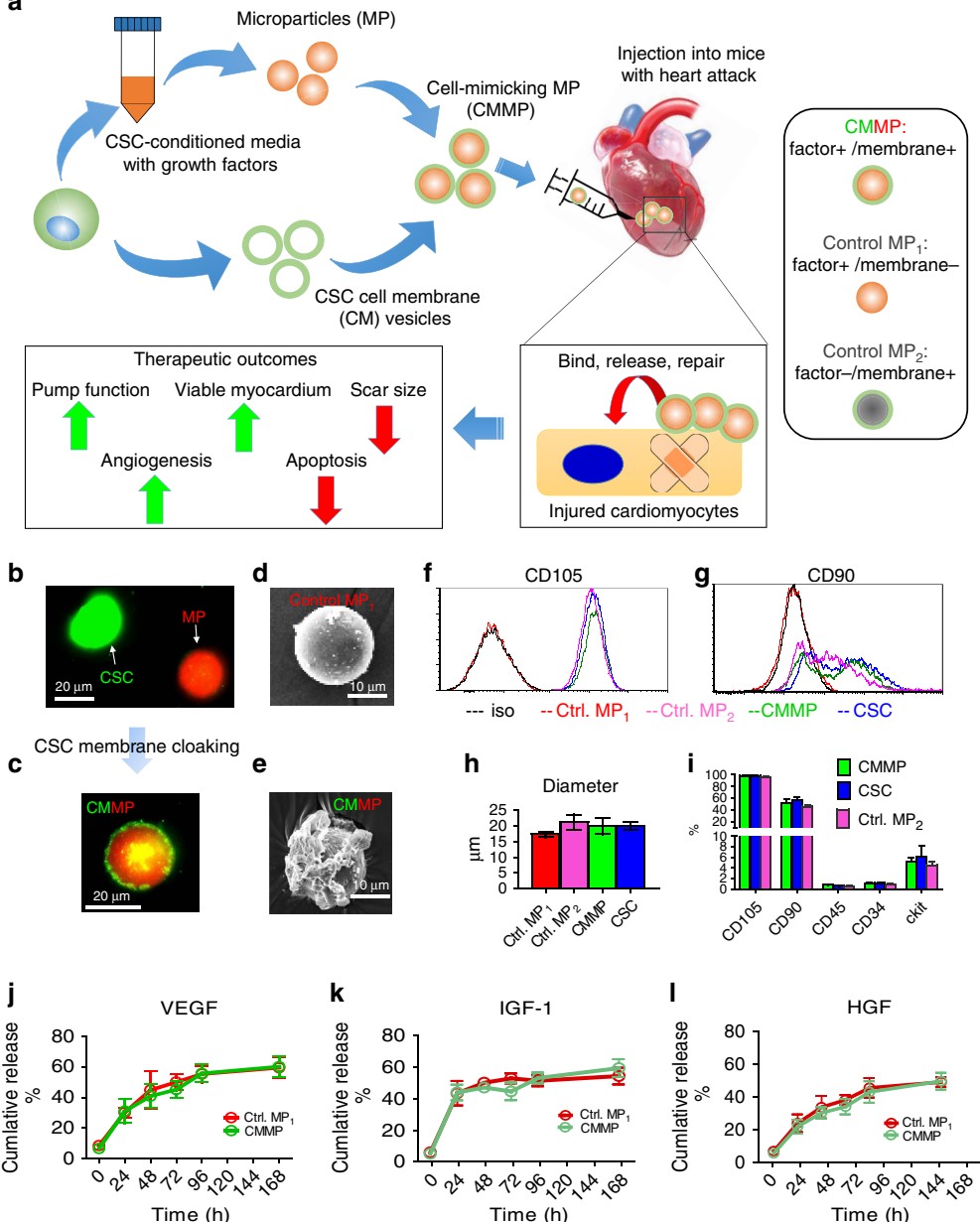

**Figure 1 | Physiochemical and biological properties of CMMPs.** (**a**) Overall biochemical design and study model of CMMPs. MPs (that is, Control MP$_1$) were fabricated from PLGA and conditioned media of human CSCs, then MPs were cloaked with membrane fragments of CSCs to form CMMPs. Control MP$_2$ was fabricated by cloaking empty PLGA particles with CSC membranes. The therapeutic potential of CMMPs was tested in a mouse model of myocardial infarction. (**b,c**) Texas red succinimidyl ester-labelled MPs (**b**, red) were cloaked with the membrane fragments of green fluorescent DiO-labelled CSCs (**b**, green) to form CMMP (**c**, red particle with green coat). Scale bar, 20 μm. (**d,e**) SEM revealed the CSC membrane fragments on CMMPs (**e**) but not on Control MP$_1$ (non-cloaked MP) (**d**). Scale bar, 10 μm. (**f,g**) Major human CSC markers CD105 (**f**) and CD90 (**g**) were positive on CMMPs and Control MP$_2$ but not on non-cloaked Control MP$_1$, indicating the successful membrane cloaking on CMMPs. (**h**) CMMPs, Control MP$_1$ and Control MP$_2$ have similar sizes to those of CSCs. $n = 3$ for each group. (**i**) CMMPs and Control MP$_2$ carried similar surface antigens as CSCs did. $n = 3$ for each group. (**j–l**) Similar release profile of CSC factors (namely vascular endothelial growth factor (VEGF), insulin-like growth factor (IGF)-1 and hepatocyte growth factor (HGF)) was observed in CMMPs and Control MP$_1$, indicating membrane cloaking did not affect the release of CSC factors from CMMPs and Control MP$_1$. $n = 3$ for each time point. All data are mean ± s.d. Comparisons between any two groups were performed using two-tailed unpaired Student's $t$-test. Comparisons among more than two groups were performed using one-way ANOVA followed by *post hoc* Bonferroni test.

stem cell biointerfacing during regeneration. This particle, named cell-mimicking MP (CMMP), contained control-released stem cell factors in its polymeric core and was cloaked with stem cell membrane fragments on the surface. Our hypothesis is that CMMP can exert similar regenerative outcomes as real CSCs but are superior to the later since they are more stable during storage and do not stimulate T-cell immune reaction since they are not real cells.

In the present study, we report for the first time a poylmer MP which emulates CSC functions during tissue repair. In a mouse model of myocardial infarction, injection of CMMPs led to preservation of viable myocardium and augmentation of cardiac functions similar to CSC therapy. CMMPs (derived from human cells) did not stimulate T-cell infiltration in immuno-competent mice, suggesting their excellent safety profile. Although our first application targeted the heart, the CMMP strategy represents a platform technology that can be applied to multiple stem cell types and the repair of various organ systems.

## Results

**Physiochemical and biological properties of CMMPs.** The biochemical design and work model of CMMPs were outlined in Fig. 1a. Briefly, Control $MP_1$ were fabricated from poly(lactic-co-glycolic acid) (PLGA) and conditioned media of human CSCs which were isolated from human hearts using the cardiosphere method as previously described[13,14] (Supplementary Fig. 1). The conditioned media contains various growth factors secreted by CSCs[10]. CSCs have been tested and proven safe and effective in Phase I/II clinical trials[1–3]. After that, MPs (Texas red

succinimidyl ester-labelled; Fig. 1b, red) were cloaked with the membrane fragments of CSCs (green fluorescent DiO-labelled; Fig. 1b, green) to become the final product CMMP (Fig. 1c, red particle with green coat). Fluorescent imaging revealed there is no specific DiO outer layer fluorescence on Texas red succinimidyl ester-labelled MPs (Control $MP_1$) after 30 min co-culture (Supplementary Fig. 2). Scanning electron microscopy (SEM) revealed the effective CSC membrane cloaking on CMMPs (Fig. 1e) but not on non-cloaked MPs (Control $MP_1$; Fig. 1d). As another control particle, Control $MP_2$ was fabricated by cloaking empty PLGA particles with CSC membranes. We fabricated CMMPs, Control $MP_1$ and Control $MP_2$ with sizes similar to those of real CSCs (Fig. 1h). As an indicator of successful membrane cloaking, flow cytometry analysis confirmed the expression of major human CSC markers (for example, CD105, CD90) on CMMPs and Control $MP_2$ but not on Control $MP_1$ (Figs 1f,g and 2). Overall, both CMMPs and Control $MP_2$ carried similar surface antigens as CSCs did (Fig. 1i). Membrane cloaking did not affect the release of CSC factors (namely vascular endothelial growth factor, insulin-like growth factor-1 and hepatocyte growth factor) from CMMPs and Control $MP_1$ (Fig. 1j–l; Supplementary Fig. 3). Snap freezing in $-80\,°C$ and thawing in water did not affect the membrane coating (Supplementary Fig. 4a), size (Supplementary Fig. 4b,c) or surface antigen expression of CMMPs (Supplementary Fig. 4d–f). These results confirmed CMMPs recapitulated the secretome and surface antigen profile of genuine CSCs. In contrast, Control $MP_1$ contained CSC secretome but not the membranes of CSCs, while Control $MP_2$ carried the membranes of CSCs successfully.

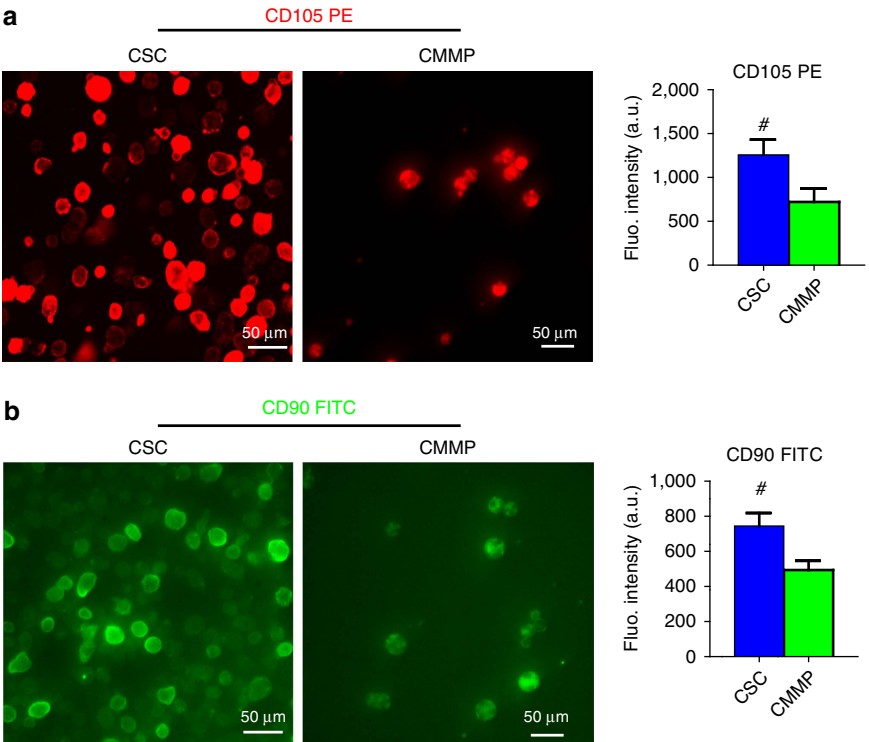

**Figure 2 | Fluorescence densities of CD105 and CD90 expressions on CMMPs and CSCs.** (**a**) Fluorescent images of CSCs (left panel) and CMMPs (right panel) labelled with CD105-PE conjugated antibody. Quantitative analysis of fluorescent intensities of CSCs (blue bar) and CMMPs (green bar). (**b**) Fluorescent images of CSCs (left panel) and CMMPs (right panel) labelled with CD90-FITC conjugated antibody. Quantitative analysis of fluorescent intensities of CSCs (blue bar) and CMMPs (green bar). $n = 6$ for each group. All data are mean ± s.d. # indicates $P < 0.05$ when compared with CMMP group. Comparisons were performed by two-tailed unpaired Student's $t$-test.

**CMMPs promote cardiomyocyte functions *in vitro*.** An important potency indicator of CSCs is their ability to promote the functions of *in vitro*-cultured cardiomyocytes. CMMPs, Control MP$_1$, Control MP$_2$, or CSCs (red, Fig. 3a) were co-cultured with neonatal rat cardiomyocytes (NRCMs; stained for alpha-sarcomeric actin (green), Fig. 3a) in plain Iscove's modified Dulbecco's medium. Solitary NRCM culture was included as the negative control. While Control MP$_1$ (red bar, Fig. 3b) increased the numbers of NRCMs as compared with those from Control MP$_2$ (pink bar, Fig. 3b) or solitary NRCM culture (white bar, Fig. 3b), the greatest NRCM numbers were seen in those co-cultured with CMMPs (green bar, Fig. 3b) and genuine CSCs (blue bar, Fig. 3b). Furthermore, CMMPs and Control MP$_1$

robustly promoted NRCM contractility (Fig. 3c) and proliferation (as indicated by Ki67-positive nuclei, Fig. 3d). Both CMMPs and Control MP$_2$ could firmly bind to cardiomyocytes, as cells did, while most non-cloaked Control MP$_1$ floated in the media (Fig. 3e). Such binding was confirmed by CMMPs' synchronized movement with adjacent beating cardiomyocytes (Fig. 3f; Supplementary Movies 1 and 2). Moreover, time-lapse imaging revealed the rolling (Fig. 2g; Supplementary Movie 3) and travelling (Fig. 3h; Supplementary Movie 4) of CMMPs on attached cardiomyocytes, suggesting the biointerfacing between CMMPs and cardiomyocytes. Such dynamic activities were not seen in non-cloaked Control MP$_1$. These *in vitro* cell-based assays suggest the regenerative potential of CMMPs in the heart.

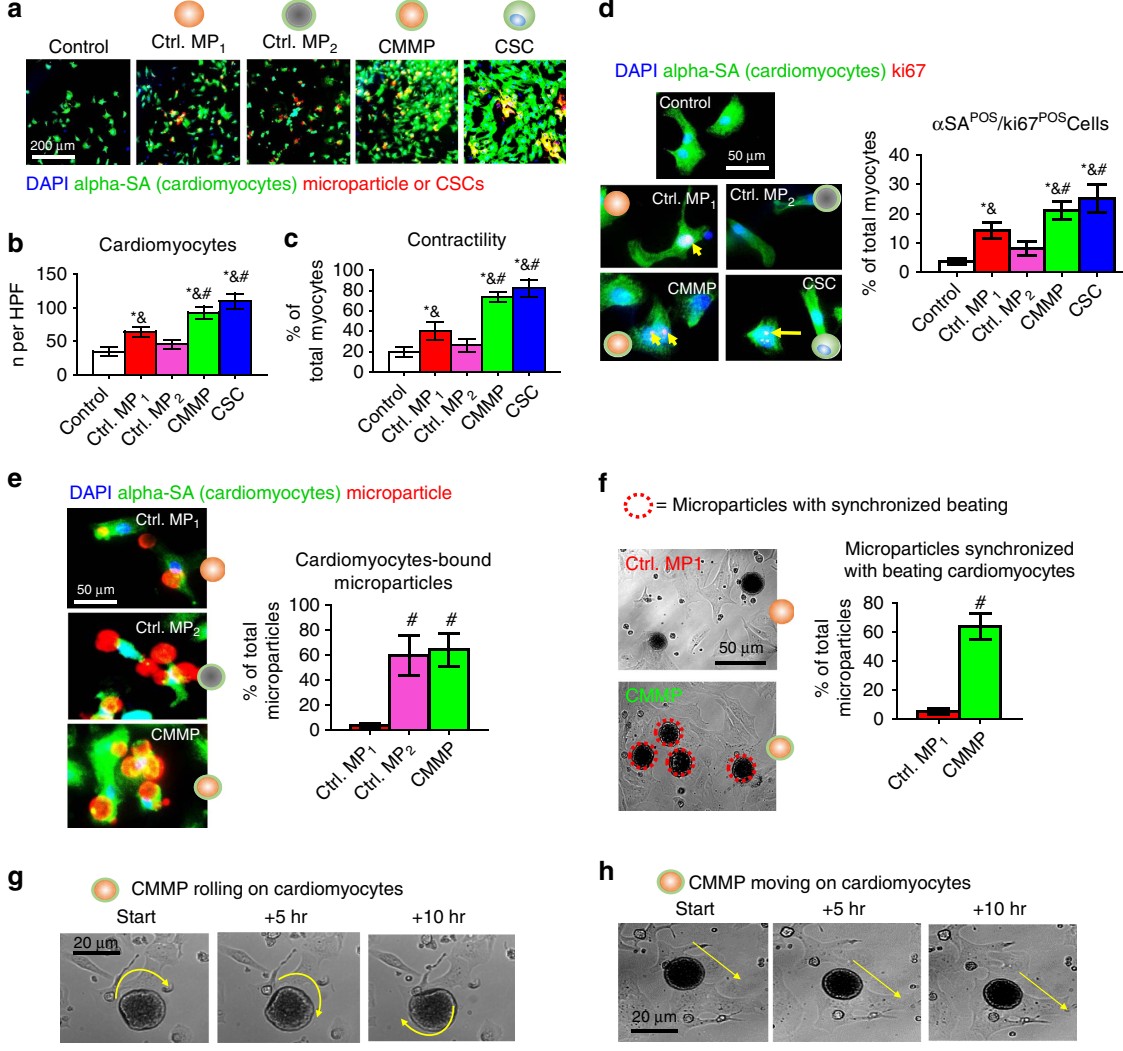

**Figure 3 | Effects of CMMPs on NRCMs functions *in vitro*.** (**a**) Representative images of cardiomyocytes stained with alpha sarcomeric actin (green) co-cultured with Control MP$_1$, Control MP$_2$, CMMPs or CSCs (red). Scale bar, 200 μm. (**b**) Quantitative analysis reflected that Control MP$_1$ (red bar) increased the numbers of NRCMs as compared with those from Control MP$_2$ (pink bar) or solitary NRCM culture (white bar), but the greatest NRCM numbers were seen in those co-cultured with CMMPs (green bar) and genuine CSCs (blue bar). $n = 5$ for each group. (**c**) Higher NRCM contractility was seen in those cultured with CMMPs (green bar) and CSCs (blue bar) compared with those cultured with Control MP$_1$ (red bar). $n = 5$ for each group. (**d**) Representative images and quantitative analysis of NRCMs stained with alpha sarcomeric actin (green) and proliferation marker Ki67 (red), treated with Control MP$_1$, Control MP$_2$, CMMPs or CSC. $n = 5$ for each group. Scale bar, 50 μm. (**e**) Representative images and quantitative analysis of CMMP (red) or Control MP$_1$ (red), Control MP$_2$ (red) binding to NRCMs (green). $n = 3$ for each group. Scale bar, 50 μm. (**f**) Representative movie screenshots and quantitation of Control MP$_1$'s and CMMP's synchronized movement with adjacent beating cardiomyocytes. $n = 3$ for each group. Scale bar, 50 μm. (**g,h**) Time-lapse imaging revealed the rolling (**g**) and travelling (**h**) of CMMPs on attached cardiomyocytes. Yellow arrows indicated the rolling or moving directions. Scale bar, 20 μm. All data are mean ± s.d. * indicates $P < 0.05$ when compared with Control group; # indicates $P < 0.05$ when compared with Control MP$_1$ group; & indicated $P < 0.05$ when compared with Control MP$_2$. Comparisons between any two groups were performed using two-tailed unpaired Student's *t*-test. Comparisons among more than two groups were performed using one-way ANOVA followed by *post hoc* Bonferroni test.

**CMMP therapy in immunodeficient mice with heart attack**. To test the therapeutic potential of CMMPs, we employed a mouse model of myocardial infarction (heart attack) by left anterior descending artery (LAD) ligation (Fig. 4a). CMMPs or Control $MP_1$ were intramyocardially injected immediately after LAD ligation. Negative or positive control animals received injection of vehicle (PBS) or CSCs, respectively. *Ex vivo* fluorescent imaging at Day 3 revealed that more CMMPs were retained in the heart after injection than Control $MP_1$ (Fig. 4b) were. This was further confirmed by histology (Fig. 4c). This was consistent with CMMP's superior binding to cardiomyocytes *in vitro* (as seen in Fig. 3). In addition, *ex vivo* fluorescent imaging indicated that the majority of CMMPs remained in the heart after injection, while 'washed away' CMMP signal could be found in the lung and the liver (Supplementary Fig. 5), consistent with the notion that the needle injection can cause vessel damage and the venous drainage brings the particles to the lungs[15]. The off-target expression in the liver may represent the leakage of CMMPs into the LV cavity during injection. Nevertheless, the majority of CMMPs remain in the heart after injection.

*In vivo* degradation of CMMPs was evident as only a negligible amount of particles remained in the heart at Day 28 (Supplementary Fig. 6). A cohort of animals was killed at Day 7 for assessment of myocardial tissue apoptosis and infiltration of macrophages in CMMP-treated animals. TdT-mediated dUTP nick end labelling (TUNEL) staining revealed the anti-apoptosis effects of CMMP: less apoptotic nuclei were detected in areas with the presence of CMMPs (green nuclei, Fig. 4d). CMMP treatment did not cause the exacerbation of inflammation: the tissue densities of CD45-positive cells were indistinguishable in areas with or without CMMPs (Fig. 4e). Masson's trichrome staining 4 weeks after treatment (Fig. 4f; red = healthy myocardium and blue = scar tissue) revealed Control $MP_1$ treatment (red bars, Fig. 4g–i) exhibited a certain degree of heart morphology protection compared with Control PBS injections (white bars, Fig. 4g–i). However, the greatest protective effects were seen in the animals treated with CMMPs (green bars, Fig. 4g–i). Such protective effects were similar to those injected with CSCs (blue bars, Fig. 4g–i). The *bona fide* efficacy indicator for stem cell therapy is the ability to ameliorate ventricular dysfunction or even boost cardiac function over time, gauged by echocardiography. Left ventricular ejection fractions (LVEFs) were measured at baseline (4 h post infarct) and 4 weeks afterwards. LVEFs were indistinguishable at baseline for all groups (Fig. 4j), indicating a similar degree of initial heart injury. Over the 4 week period, the LVEFs in control (PBS or saline)-treated animals continued deteriorating (white bar, Fig. 4k) while the Control $MP_1$-treated animals exhibited a trend of LVEF augmentation (red bar, Fig. 4k) but did not reach statistical significance. CMMP treatment robustly boosted cardiac function with the highest LVEFs at 4 weeks (green bar, Fig. 4k). Such treatment effects were indistinguishable from those of CSC treatment with real CSCs (blue bars, Fig. 4k). Histological analysis indicated that such functional benefits by CMMP treatment were accompanied by remuscularization (Fig. 5a), proliferation of endogenous cardiomyocytes (Fig. 5b), augmentation of blood flow (Fig. 5c), and increase of vessel density (Fig. 5d) in the post-MI heart.

**CMMP injection does not promote T-cell infiltration in normal mice**. To evaluate the local T-cell immune response to CMMPs, immune-competent CD1 mice were intramyocardially injected with human CSCs or CMMPs. Animals were killed 7 days after injection for assessment of immune rejection in the heart, as gauged by $CD3^+$ and $CD8^+$ T cell infiltration (Fig. 6a). CMMP (red) injection elicits negligible T-cell rejection as very few $CD3^+$ (green) or $CD8^+$ (green) T cells were detected in the

heart (Fig. 6c,e). In contrast, severe rejection was detected in mouse hearts treated with human CSCs: injected CSCs (red) were surrounded by clusters of $CD3^+$ (green) or $CD8^+$ (green) T cells (Fig. 6b,d). Quantitative analysis also confirmed that CMMP stimulated negligible local T-cell infiltration as compared with the severe T-cell stimulation by human CSCs (Fig. 6f,g).

## Discussion

The last one and a half decades witnessed the booming of stem cell therapies for multiple diseases[16–18]. Deviating from the initial perspective that stem cells exert their therapeutic effects through direct cell differentiation and tissue replacement, the paradigm has shifted as emerging evidence suggests that most adult stem cell types exert their beneficial effects through paracrine mechanisms (soluble factors)[19–21]. In addition, studies further suggest that cell–cell contact between the injected cells and the host cells plays an important role in tissue regeneration[11]. PLGA, as a biocompatible and biodegradable polymer, has provided a safe and non-toxic building block for various control-release systems[22]. Previous studies have demonstrated the success of coating PLGA nanoparticles with cell membranes from red blood cells[23], platelets[24] and cancer cells[25,26]. Inspired by these findings, we designed CMMPs and demonstrated the therapeutic effects of CMMPs in an experimental myocardial infarction model. The comparison between CMMP and CSCs is outlined in Supplementary Table 1. CMMP represents a synthetic MP functionalized with both stem cell membranes and secretome, harnessing the power of these two major components of stem cell-induced regeneration. Moreover, CMMP overcomes several major limitations of live stem cells as therapy products. First, stem cells need to be carefully cryo-preserved and thawed before they can be sent to the clinic. As living organisms, how the cells are prepared and processed can greatly affect the therapeutic outcomes. Second, stem cell transplantation carries certain risks (for example, tumourogenecity and immunogenicity if allogeneic or xenogeneic cells were used). CMMPs will most likely be delivered intramyocardially via direct muscle injection. Such injection normally requires open-chest surgery. However, percutaneous options are becoming available with the implementation of the NOGA mapping systems[27]. Moreover, our future studies will explore the potential of vascular delivery of CMMPs (for example, intracoronary, intravenous) with the focus on targeting CMMPs to the injury and promoting extravasation through the mechanism of angiopellosis[28,29]. One caveat of our study is that with the existing assay it is difficult to conclude whether cardiomyocytes (or their progenitors) really are proliferating and leading to remuscularization after CMMP injection. Although this proof-of-concept study targets the heart, CMMP represents a platform technology that is generalizable to other stem cell types and the repair of various other organ systems.

## Methods

**Derivation and culture of human CSCs**. Institutional review board approval was obtained for all procedures, and informed consent was achieved from all patients. Human CSCs were derived from donor human hearts as previously described[5,13]. Briefly, myocardial tissues were minced into small pieces (about 2 mm³), then washed with PBS and digested with collagenase solution (Sigma, St. Louis, MO, USA). The tissue fragments were cultured as 'cardiac explants' on plates coated with 0.5 mg ml$^{-1}$ fibronectin (Corning, Corning, NY, USA) in Iscove's modified Dulbecco's medium (Invitrogen, Carlsbad, CA, USA) supplemented with 20% fetal bovine serum (Corning), 0.5% Gentanicin (Gibco, Life Technologies, California, USA), 0.1 mM 2-mercaptoethanol (Invitrogen, Carlsbad, CA, USA) and 1% L-glutamine (Invitrogen, Carlsbad, CA, USA). Within 1–2 weeks, a layer of stromal-like flat cells, and phase-bright round cells, emerged from the cardiac explant with phase bright cells over them. These cardiac explant-derived cells were collected using TryPLE Select (Gibco), and then seeded at a density of $2 \times 10^4$ cells ml$^{-1}$ in UltraLow Attachment flasks (Corning) for cardiosphere formation. In about 1 week, explant-derived cells spontaneously aggregated into

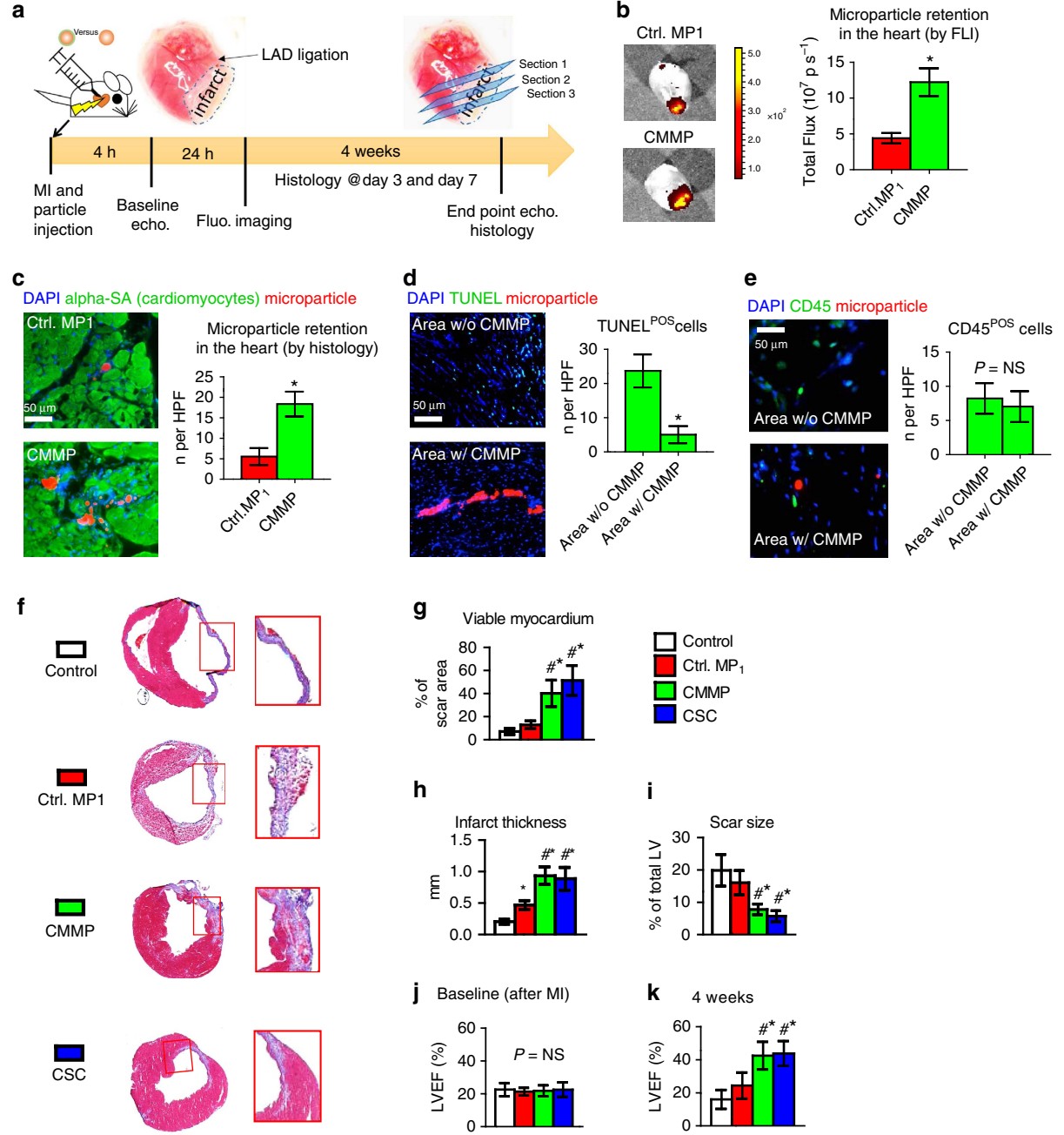

**Figure 4 | CMMPs ameliorate ventricular dysfunction and promote cardiac repair in a mouse model of heart attack.** (**a**) Schematic showing the overall design of animal experiments to test the therapeutic benefits of CMMPs in a mouse model of myocardial infarction. (**b**) Representative *ex vivo* fluorescent imaging of mouse hearts and quantitative analysis of fluorescent intensities at Day 3 after Control MP₁ or CMMP injections. $n = 3$ animals per group. * indicated $P < 0.05$ when compared with Control MP₁ group. (**c**) Representative microscopic images and quantitative analysis of mouse hearts (myocytes stained with alpha sarcomeric actin (green)) 3 days after injection of Control MP₁ (red) or CMMPs (red). $n = 3$ animals per group. Scale bar, 50 μm. * indicated $P < 0.05$ when compared with Control MP₁ group. (**d**) Representative fluorescent micrographs and quantitative analysis showing the presence of TUNEL[+] apoptotic cells (green) in CMMP-treated hearts at Day 7. $n = 3$ animals per group. Scale bar, 50 μm. * indicated $P < 0.05$. (**e**) Representative fluorescent micrographs showing the presence of CD45[+] cells (green) in the hearts treated with or without CMMPs (red) at Day 7. $n = 3$ animals per group. Scale bar, 50 μm. NS indicated $P > 0.05$. (**f**) Representative Masson's trichrome-stained myocardial sections 4 weeks after treatment with Control PBS, Control MP₁, CMMPs or CSCs. In this staining blue = scar tissue and red = viable myocardium. Snapshots = high magnification images of the red box area. (**g–i**) Quantitative analyses of viable myocardium (**g**), infarct thickness (**h**) and scar size (**i**) from the Masson's trichrome images. $n = 5$ animals per group. (**j,k**) LVEF was measured by echocardiography at baseline (4 h post-MI) and 4 weeks afterward in Control PBS, Control MP₁, CMMP and CSC groups. $n = 7$ animals per group. * indicated $P < 0.05$ when compared with Control group; # indicated $P < 0.05$ when compared with Control MP₁ group; NS indicated $P > 0.05$. All data are mean ± s.d. Comparisons between any two groups were performed using two-tailed unpaired Student's *t*-test. Comparisons among more than two groups were performed using one-way ANOVA followed by *post hoc* Bonferroni test.

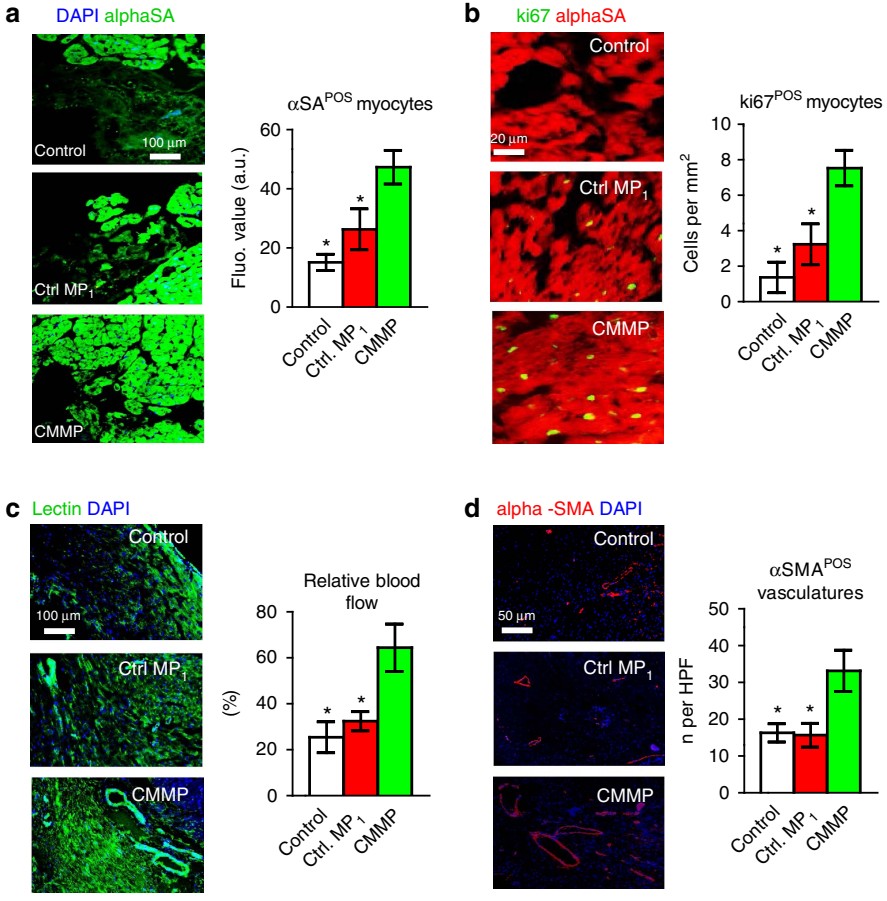

**Figure 5 | Injection of CMMPs promotes angiomyogenesis. (a)** Representative images showing alpha sarcomeric actin (αSA)-positive cardiomyocyte nuclei (green) in control PBS-, Control MP₁- or CMMP-treated hearts at 4 weeks. The numbers of αSA-positive nuclei were quantified. $n = 3$ animals per group. Scale Bar, 100 μm. **(b)** Representative images showing Ki67-positive cardiomyocyte nuclei (green) in control PBS-, Control MP₁- or CMMP-treated hearts at 4 weeks. The numbers of Ki67-positive nuclei were quantified. $n = 3$ animals per group. Scale Bar, 20 μm. **(c)** Representative images showing lectin-labelled blood vessels (green) in control PBS- and Control MP₁- or CMMP-treated hearts at 4 weeks. The lectin fluorescent intensities were quantified. $n = 3$ animals per group. Scale Bar, 100 μm. **(d)** Representative images showing arterioles stained with alpha smooth muscle actin (αSMA, red) in control PBS-, Control MP₁- or CMMP-treated hearts at 4 weeks. The numbers of αSMA positive vasculatures were quantified. $n = 3$ animals per group. Scale Bar, 50 μm. * indicates $P < 0.05$ when compared with CMMP group. All data are mean ± s.d. Comparisons among more than two groups were performed using one-way ANOVA followed by *post hoc* Bonferroni test.

cardiospheres. Cardisophere-derived CSCs were generated by seeding collected cardiospheres on fibronectin-coated plates. All cultures were incubated in 5% $CO_2$ at 37 °C.

**Fabrication of Control MPs and CMMPs.** CSC factor-loaded PLGA microparticles (Control MP₁) were fabricated by a water/oil/water (w/o/w) emulsion technique. Briefly, human CSC conditioned media as the internal aqueous phase with polyvinyal alcohol (0.1% w/v) was mixed in methylene chloride (DCM) containing PLGA as the oil phase. The mixture was then sonicated on ice for 30 s using a sonicator with a Microtip probe (Misonix, XL2020, Farmingdale, NY, USA). After that, the primary emulsion was immediately introduced into water with polyvinyl alcohol (0.7% w/v) to produce a w/o/w emulsion. The secondary emulsion was emulsified for 5 min on a high-speed homogenizer. The w/o/w emulsion was continuously stirred overnight at room temperature to promote solvent evaporation. The solidified MPs, namely Control MP₁, were then centrifuged, washed three times with water, lyophilized and stored at − 80 °C. To prepare CMMPs, DiO (Invitrogen)-labelled CSCs went through three freeze/thaw cycles. After which, the disrupted CSCs were sonicated for ∼5 min at room temperature along with the Control MP₁. After that, the particles were washed three times in PBS by centrifugation. Control MP₂ was fabricated by cloaking empty PLGA particles with CSC membranes. Successful membrane coating was confirmed using fluorescent microscopy.

**Protein release studies.** Total protein contents in MPs were determined using the following method. Approximately 10 mg freeze-dried microparticles were dissolved in 1 ml DCM for 60 min. Then, 1 ml PBS was added into solution followed by agitation for 10 min to extract protein from DCM into PBS. After centrifugation,

the concentration of protein in the aqueous phase was determined by a BCA Protein Assay Kit (Thermo Fisher Scientific, Waltham, MA, USA). For release studies, MPs were incubated in PBS at 37 °C. Supernatant was collected at various time points and the concentrations of proteins were determined by commercially available ELISA kits (R & D Systems, Minneapolis, MN, USA)

**Scanning electron microscopy.** The morphology of microparticles was studied by SEM (Philips XL30 scanning microscope, Philips, The Netherlands). Freeze-dried microparticles were mounted on aluminium stubs with double-sided tape and coated with a thin layer of gold. The coated specimen was then scanned and photographed under the microscope at an acceleration voltage of 15 kV.

**Flow cytometry.** To characterize the phenotypes of Control MP₁, Control MP₂, CMMP, and CSC, flow cytometry was performed using a CytoFLEX Flow Cytometer (Beckman Coulter, Brea, CA) and analysed using FCS Express software (De Novo Software, Los Angeles, CA). Briefly, cells were incubated with FITC, PE, or APC-conjugated antibodies against CD105 (10 μl per 40 μl of sample, FAB 10971R, R&D Systems), CD90 (10 μl per 40 μl of sample, BD 555595), CD45 (10 μl per 40 μl of sample, BD 555482), CD34 (10 μl per 40 μl of sample, BD 555821) and CD117 (5 μl per 45 μl of sample, c-kit; BD 550412) from BD company (Franklin Lakes, NJ) for 60 min. Isotype-identical antibodies from BD Company served as negative controls.

**Immunocytochemistry.** Control MP₁, Control MP₂, CMMP, and CSC were pre-labelled with red-fluorescent Texas red succinimidyl ester (1 mg ml⁻¹ (Invitrogen, Carlsbad, California)). NRCM or NRCM co-cultured with pre-labelled

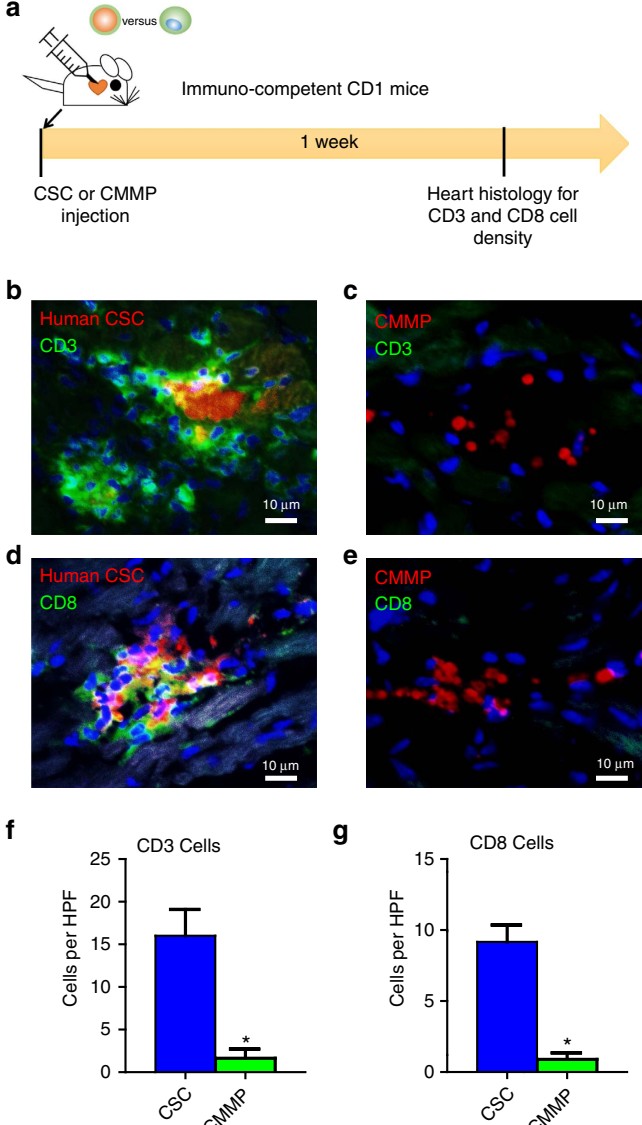

**Figure 6 | CMMP injection does not stimulate local T-cell immune response in immunocompetent mice.** (**a**) Schematic showing the overall animal study design to assess the local T-cell immune reaction induced by human CSCs or CMMPs derived from human CSCs. (**b,c**) Representative fluorescent images showing the presence of infiltrated CD3$^+$ T cells (green) in CSCs (red, b)- or CMMPs (red, c)-injected hearts at Day 7. Scale bar, 10 μm. (**d,e**) Representative fluorescent images showing the presence of infiltrated CD8$^+$ T cells (green) in CSCs (red, **b**) - or CMMPs (red, **c**)-injected hearts at Day 7. Scale bar, 10 μm. (**f,g**) Quantitative analysis of CD3$^+$ and CD8$^+$ T cells in CSCs (blue bar)- or CMMPs (green bar)-injected hearts at Day 7. $n = 3$ animals per group. All data are mean ± s.d. Comparisons between any two groups were performed using two-tailed unpaired Student's $t$-test. * Indicated $P < 0.05$ when compared with CSC group.

Control MP$_1$, Control MP$_2$, CMMP, and CSC were plated onto fibronectin-coated chamber slides (BD Biosciences) and subsequently fixed with 4% paraformaldehyde before immunocytochemistry (ICC) staining. Slides were stained with the antibodies against α-SA (1:100, a7811, Sigma) or ki67 (1:100, ab15580, Abcam) and detected by FITC- or Texas Red-conjugated secondary antibodies (1:100). Nuclei were stained with DAPI. Images were taken with an epi-fluorescent microscope (Olympus IX81).

**Mouse model of myocardial infarction.** All animal work was compliant with the Institutional Animal Care and Use Committee at North Carolina State University.

The method to induce myocardial infarction in mice was based on previous studies[30]. Briefly, male SCID Beige mice were anaesthetized with 3% isofluorane combined with 2% oxygen inhalation. Under sterile conditions, the heart was exposed by a minimally invasive left thoracotomy and acute myocardial infarction (AMI) was produced by permanent ligation of the LAD coronary artery. Immediately after AMI induction, the heart was randomized to receive one of the following four treatment arms: (1) 'Control (PBS)' group: intramyocardial injection of 50 μl PBS into the heart immediately after AMI; (2) 'Control MP$_1$' group: intramyocardial injection of $1 \times 10^5$ Control MP$_1$ in 50 μl PBS into the heart immediately after AMI; (3) 'CMMP' group: intramyocardial injection of $1 \times 10^5$ CMMPs in 50 μl PBS into the heart immediately after AMI; (4) 'CSC' group: intramyocardial injection of $1 \times 10^5$ CSCs in 50 μl PBS into the heart immediately after AMI. To enable visualization of Control MP$_1$ or CMMP in a cohort of animals, we pre-labelled the Control MP$_1$ or CMMP with Texas Red-X succinimidyl ester (1 mg ml$^{-1}$ (Invitrogen, Carlsbad, California)).

**Ex vivo fluorescent imaging for biodistribution of CMMPs.** Seven days after injection, a cohort of mice receiving CMMPs were killed; their heart, lung, spleen, liver, and kidneys were removed for biodistribution studies. *Ex vivo* fluorescent imaging was performed with an IVIS Xenogen *In Vivo* Imager (Caliper Life-sciences, Waltham, MA).

**Heart morphometry.** After the echocardiography study at 4 weeks, all animals were killed and hearts were collected and frozen in optimum cutting temperature (OCT) compound. Specimens were sectioned at 10 μm thickness from the apex to the ligation level with 100 μm intervals. Masson's trichrome staining was performed as described by the manufacturer's instructions (HT15 Trichrome Staining (Masson) Kit; Sigma-Aldrich). Images were acquired with a PathScan Enabler IV slide scanner (Advanced Imaging Concepts, Princeton, NJ). From the Masson's trichrome stained images, morphometric parameters including viable myocardium, infarct thickness and scar size were measured in each section with NIH ImageJ software. The percentage of viable myocardium as a fraction of the scar area (infarcted size) was quantified as described[31–33]. Three selected sections were quantified for each animal.

**Cardiac function assessment.** All animals underwent transthoracic echocardiography under 1.5% isofluorane-oxygen mixture anaesthesia in supine position at 4 h and 4 weeks. The procedure was performed by an animal cardiologist blind to the experimental design using a Philips CX30 ultrasound system coupled with an L15 high-frequency probe. Hearts were imaged in 2D in long-axis views at the level of the greatest LV diameter. EF was determined by using the formula (LVEDV–LVESV/LVEDV) × 100%.

**Histology.** For immunohistochemistry staining, heart cryosections were fixed with 4% paraformaldehyde, permeabilized and blocked with Protein Block Solution (DAKO, Carpinteria, CA) containing 0.1% saponin (Sigma, St Louis, MO), and then incubated with the following antibodies overnight at 4 °C: mouse anti-alpha sarcomeric actin (1:100, a7811, Sigma), rabbit anti-CD45 (1:100, ab10559, Abcam, Cambridge, United Kingdom), mouse anti-Actin, α-Smooth Muscle antibody (1:100, A5228, Sigma), rabbit anti-Ki67 (1:100, ab15580, Abcam), rabbit anti-CD3 (1:100, ab16669, Abcam) and mouse anti-CD8 alpha (1:100, mca48r, abd Serotec, Raleigh, NC ). FITC- or Texas-Red secondary antibodies (1:100) were obtained from Abcam Company and used for the conjunction with these primary antibodies. For assessment of cell apoptosis, heart cryosections were incubated with TUNEL solution (Roche Diagnostics GmbH, Mannheim, Germany) and counterstained with DAPI (Life Technology, NY, USA). For assessment of angiogram, heart cryosections were incubated with Lectin (FL-1171, Vector laboratories, Burlingame, CA, USA). Images were taken by an Olympus epi-fluorescence microscopy system.

**Immunogenicity studies for human CSCs and CMMPs.** Immuno-competent male CD1 mice were anaesthetized with 3% isofluorane combined with 2% oxygen inhalation. Under sterile conditions, the heart was exposed by a minimally invasive left thoracotomy, and the heart was randomized to receive one of the two treatments: (1) 'CMMP' group: intramyocardial injection of $1 \times 10^5$ CMMPs in 50 μl PBS into the heart; (2) 'CSC' group: intramyocardial injection of $1 \times 10^5$ human CSCs in 50 μl PBS into the heart. To enable visualization of CMMPs or CSCs, they were pre-labelled with red fluorophore.

**Statistical analysis.** All results are expressed as mean ± s.d. Comparison between two groups was performed with two-tailed Student's $t$-test. Comparisons among more than two groups were performed using one-way ANOVA followed by *post hoc* Bonferroni test. Differences were considered statistically significant when the $P$ value < 0.05.

**Data availability.** The authors declare that all data supporting the findings of this study are available within the article and its Supplementary Information files or from the corresponding author on reasonable request.

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

## Acknowledgements

This work was supported by funding from National Institute of Health HL123920, NC State University Chancellor's Faculty Excellence Program, NC State Chancellor's Innovation Fund, University of North Carolina General Assembly Research Opportunities Initiative grant and National Natural Science Foundation of China 81370216, 81570274, 31670895 and U1404802, Science and Technology Innovation Team Support Project of Henan Province 14IRTSTHN018, Innovation Team of Science and Technology Project of Henan Province 164200510012. J.T. is supported by China Scholarship Council. The study is also partially supported by the Cooperative Research Grant(s) of Atomic Bomb Disease Institute at Nagasaki University, Japan.

## Author contributions

J.T., D.S., T.G.C., J.Z., Q.K. and K.C. designed research, performed biochemical, cellular and animal experiments, analysed the data and drafted the paper. Z.W., T.A.A., A.C.V., M.T.H., P.-U.D. and J.C. performed cellular and *in vitro* experiments. T.-S.L., J.Z., Q.K., and K.C. directed the research and provided financial support.

## Additional information

**Competing financial interests:** The authors declare no competing financial interests.

