## [Peer Review File · Nature Communications]

Reviewer #1 (Remarks to the Author):

This study summarizes the development of a novel and biomimetic particle technology that consists of a biodegradable polymeric core encapsulating cellularly derived growth factors and encased with membrane-fragments of cardiac stem cells. This technology was demonstrated to increase cardiac cell viability and function in vitro as well as increasing post-myocardial infarct tissue repair in vivo. Strikingly, the authors also demonstrated how their biomimetic particles could extravasate after being administered intravenously. The novelty and importance of these findings makes this an appropriate submission for Nature Communications. However, there are a few issues that should be addressed prior to publication:

- One control seems to be missing from all of the studies presented in the paper. That control would be the empty PLGA microparticle, coated with the membrane derived from the CSCs. This would be a very interesting condition to see in the studies presented, as it would help elucidate the mechanism of action of these particles. Ideally this control would be included in all the studies but most importantly the in vitro experiments.
- How translatable would intravenous administration be to a mammalian system? The particles are very large (~15-20µm) which is much larger than the width of capillaries. Furthermore, the polymeric core of PLGA as synthesized by the authors would be very rigid and would not be able to squeeze through capillaries like blood cells do. Therefore there is a concern that if administered intravenously, these particles would clog capillaries and smaller blood vessels, particularly in the lungs. A biodistribution experiment that tracks the distribution of the particles in the blood and organs in a mouse model would be key to support the intravenous potential of these particles.
- For visualization of the particle membranes, the authors utilized DiO. This dye is known to be very lipophilic and therefore has the potential to stick to the hydrophobic PLGA surface to give the false illusion of successful membrane coating. A confocal visualization of an antibody stain or a control experiment where DiO is incubated with particles would be useful to negate this concern.
- Although the application and the specific formulation are very novel, the membrane coating procedure is not. In recent years different groups have shown that one can coat PLGA particles with red blood cell membranes, platelets, leukocytes, and cancer cell membranes. It would be good if the authors could reference these papers in this manuscript and incorporate them into their discussion.
- There is a misspelling of the word "flask" in the Methods section line 241.

Reviewer #2 (Remarks to the Author):

The manuscript by Tang et al. describes an approach to gain some of the benefits of stem cell therapy without some of the complexities, using conditioned media from stem cells and membrane components from stem cells on a micro particle platform. The controls to compare conditioned media micro particles alone versus with additional membrane component coating, moreover compared to stem cell therapy, are very well done. The results are interesting and definitive.

1. Line 63: "did not elicit immune response" is perhaps overstated, in that the investigators only looked for local T cell infiltration. The authors should let be more precise.
2. Line 74: The sentence stating that stem cells act (mainly, or even only) by paracrine effects may be over-stated. The authors should state simply that one mode of action can be by paracrine "mini-drug pump" effects.

3. Line 93: It is not clear exactly which paper is intended in the reference to "previously described".
4. Line 95: "medial" should be "media".
5. Line 96: "texas" should be "Texas".
6. Line 136: Here, it would be good to remind the reader that the control MPs contained the conditioned media components but not the cell membrane components.
7. Line 154: "blur" should be "blue"
8. Section starting at Line 169, including relevant discussion: Firstly, it is not clear what the relevance of the xeno'graft' experiment is. Do the investigators imagine treating humans with non-human components? Here, mice were treated with human components. Second, the authors should be more circumspect - there may have indeed been an immune response, but not one that led to local T cell infiltration.
9. Section starting at Line 181, including relevant discussion: It is an interesting observation that the membrane component-including MPs are expelled from the vasculature, however no mention is made that this is not site-specific. Were this to be done in a mouse, or a patient, a very small fraction of the injected dose would be expelled in the infarct, as opposed to the rest of the vasculature. As such, discussion is necessary to indicate that additional targeting would be required to make this make sense.

Reviewer #3 (Remarks to the Author):

The article by Tang et al., seeks to determine whether PLGA microparticles loaded with secreted proteins and membranes from cardiac stem cells provide an easy to handle, efficacious treatment in mice induced to have myocardial infarction by coronary ligation. Basic analysis is done to show the marker profile and secreted growth factor profile of the MP (loaded with CSC conditioned medium) and CMMP (loaded with CSC conditioned medium and membranes) is the same. They also show CMMPs elicit more response when cultured with neonatal rat cardiomyocytes in vitro. Translation in vivo shows CMMPs allow a similar level of recovery on MI mice when compared to CSC grafts; both CMMP and CSC transplants perform better than MP alone. Finally, the authors use a zebrafish model to show that CMMPs but not MPs can escape the circulation, arguing that this indicates that intravenous delivery of the CMMPs is feasible.

Overall, this might be an interesting study but currently it is not particularly easy to follow and is incomplete. Here are some suggestions for clarification and improvement:

- 1) The authors state that "CMMPs carried the same secreted proteins and membranes as genuine cardiac stem cells did". However, the data presented look at only a limited set of surface markers (CD105, CD90, CD45, CD34, ckit) and growth factors (VEGF, IGF1, HGF). The same is true of the statement "MPs contained CSC secretome but not the membrane of CSCs" but only 3 growth factors were assessed. Therefore, these conclusions are not substantiated.
- 2) Related to (1), the authors have investigated only the percent positive CMMPs relative to CSCs. There are no data presented on fluorescence intensity, which would be an indication of density of markers and how this compares between the CMMPs and CSCs.
- 3) in Fig. 1j-l, release profiles are given for VRGF, IGF1 and HGF. This is cited as % release. It would be better to provide details of quantity per particle and compare MP, CMMP and CSCs - currently, there are no data on CSCs, which makes it difficult to evaluate how these particles compare with 'real' cells
- 4) I am confused by the claim that "CMMP... will not elicit immune reactions since they are not real cells". This is factually incorrect since the CMMPs are coated with proteins, sugars and lipids, all of which can readily elicit an immune response.
- 5) The purpose of Fig 2 is confusing. This only shows that CMMPs interact more effectively with NRCMs than do MPs. The aim of the study seems to be to show that CMMPs are at least as good if not better than CSCs, so why are CSCs not included in the experiments?

6) The statement "In vivo degradation of CMMPs were evident as only negligible amount of particles remained in the heart at Day 28 (Supplementray Fig. 3)." Not necessarily correct. The authors do not appear to have checked any other organ systems. It is most likely the CMMPs have been washed away to end up in the spleen, thymus, liver etc. The authors should check these organs to identify distribution and cross compare with animals treated with CSCs

7) The analysis of the animal hearts is rather superficial. For example, the reader is told only that there is viable myocardium of scar tissue in Fig. 3. There is no information on potential mechanism of action, e.g. is there remuscularisation to account for the large differences in ventricular wall thickness? If this effect is anti-apoptotic, some insight into the protective nature is needed. Also, if CMMPs are more stable, easier to cryopreserve and transport than CSCs, there need to be convincing data that show this over either a time course of storage or at least no cryo vs cryo.

8) The purpose of Figure 4 is not clear. This shows in a zebrafish model that CMMPs can escape through the vasculature. However, there is no biological significance to this experiment. Can CMMPs home to the heart or do they become spread throughout the various organ systems in the body? Is this further improved by localised delivery to the vasculature of the heart? These questions need a detailed investigation that would be better saved for a separate manuscript. The authors should focus on solidifying whether CMMPs can improve heart function by direct injection and of they can what is the mechanism of action

Minor:

- The text and fig 3a say LVEF are measured after 4 weeks but fig 3k says 3 weeks
- There are typographical errors throughout the manuscript

RESPONSE

Reviewer 1

We thank the reviewer for her/his constructive comments which have allowed us to perform the new experiments to improve the study.

- 1. One control seems to be missing from all of the studies presented in the paper. That control would be the empty PLGA microparticle, coated with the membrane derived from the CSCs. This would be a very interesting condition to see in the studies presented, as it would help elucidate the mechanism of action of these particles. Ideally this control would be included in all the studies but most importantly the in vitro experiments.**

Re: We agree with the reviewer. We have included a new Control, namely Control MP₂, which are empty PLGA particles with CSC membrane coating but without CSC factors. This control is included in the in vitro experiments in Fig. 2. We found that Control MP₂ fails to promote cardiomyocytes functions, suggesting CSC factors play the main role in augmenting cardiomyocytes functions, while CSC membranes synergize such beneficial effects by promoting the adhesion of CMMPs on myocytes.

- 2. How translatable would intravenous administration be to a mammalian system? The particles are very large (~15-20µm) which is much larger than the width of capillaries. Furthermore, the polymeric core of PLGA as synthesized by the authors would be very rigid and would not be able to squeeze through capillaries like blood cells do. Therefore there is a concern that if administered intravenously, these particles would clog capillaries and smaller blood vessels, particularly in the lungs. A biodistribution experiment that tracks the distribution of the particles in the blood and organs in a mouse model would be key to support the intravenous potential of these particles.**

Re: Indeed, other reviewers/editors have also pointed out the premature nature of the zebrafish data. Per the editor's suggestion, we have removed the zebrafish data. Nevertheless, we have added Discussion regarding the possible delivery routes for CMMPs.

On Page 10 Line 12 (Discussion Part), we added: "CMMPs will most likely be delivered intramyocardially via direct muscle injection. Such injection normally requires open-chest surgery. However, percutaneous options are becoming available with the implementation of the NOGA mapping systems (Gyöngyösi M, et al. *Nat Rev Cardiol.* 8:393-404 (2011)). Moreover, our future studies will explore the potential of vascular delivery CMMPs (e.g. intracoronary, intravenous) with the focus on targeting CMMPs to the injury and promoting extravasation."

- 3. For visualization of the particle membranes, the authors utilized DiO. This dye is known to be very lipophilic and therefore has the potential to stick to the hydrophobic PLGA surface to give the false illusion of successful membrane coating. A confocal visualization of an antibody**

stain or a control experiment where DiO is incubated with particles would be useful to negate this concern.

Re: This is an excellent point. After the fabrication process, the CMMPs underwent extensive washes in PBS to eliminate nonspecific binding of DiO. In addition, flow cytometry analysis (Fig. 1i) indicated the presence of major CSC markers on CMMPs, which wouldn't be possible if it is nonspecific DiO binding. Nevertheless, we have included a new control experiment in this revision where DiO is incubated with MP particles (Control MP₁) (Suppl fig. 2). Apart from some background green fluorescence, there is no specific DiO fluorescence on the particles.

4. Although the application and the specific formulation are very novel, the membrane coating procedure is not. In recent years different groups have shown that one can coat PLGA particles with red blood cell membranes, platelets, leukocytes, and cancer cell membranes. It would be good if the authors could reference these papers in this manuscript and incorporate them into their discussion

Re: The reviewer's comments are well taken: our work has been benefitted from many prior work on membrane coating. We have added new references (Refs. 22 - 26) for these papers and included them into the Discussion.

5. There is a misspelling of the word "flask" in the Methods section line 241.

Re: We apologize for this typo. A correction has been made in the revised manuscript.

Reviewer 2

We thank the reviewer for her/his constructive comments which have allowed us to perform the new experiments to improve the study.

1. Line 63: "did not elicit immune response" is perhaps overstated, in that the investigators only looked for local T cell infiltration. The authors should let be more precise.

Re: The reviewer is correct. We have changed the statement to “did not stimulate local T cell infiltration”.

2. Line 74: The sentence stating that stem cells act (mainly, or even only) by paracrine effects may be over-stated. The authors should state simply that one mode of action can be by paracrine "mini-drug pump" effects.

Re: The reviewer is correct. We have changed the statement to “one important mode of therapeutic action is the secretion of paracrine factors by injected stem cells that act like “mini-drug pumps” to promote endogenous repair”.

3. **Line 93: It is not clear exactly which paper is intended in the reference to "previously described".**

Re: We have added the reference.

4. **Line 95: "medial" should be "media"**
5. **Line 96: "texas" should be "Texas".**
6. **Line 136: Here, it would be good to remind the reader that the control MPs contained the conditioned media components but not the cell membrane components.**
7. **Line 154: "blur" should be "blue"**

Re: We have made the changes/corrections according to the reviewer's suggestions.

8. **Section starting at Line 169, including relevant discussion: Firstly, it is not clear what the relevance of the xeno'graft' experiment is. Do the investigators imagine treating humans with non-human components? Here, mice were treated with human components. Second, the authors should be more circumspect - there may have indeed been an immune response, but not one that led to local T cell infiltration.**

Re: We agree with the reviewer. In the revised paper, we have toned down our statement regarding the immune response to CMMP injections.

9. **Section starting at Line 181, including relevant discussion: It is an interesting observation that the membrane component-including MPs are expelled from the vasculature, however no mention is made that this is not site-specific. Were this to be done in a mouse, or a patient, a very small fraction of the injected dose would be expelled in the infarct, as opposed to the rest of the vasculature. As such, discussion is necessary to indicate that additional targeting would be required to make this make sense.**

Re: Indeed, other reviewers/editors have also pointed out the premature nature of the zebrafish data. Per the editor's suggestion, we have removed the zebrafish data. Nevertheless, we have added Discussion regarding the possible delivery routes for CMMPs.

On Page 10 Line 12 (Discussion Part), we added: "CMMPs will most likely be delivered intramyocardially via direct muscle injection. Such injection normally requires open-chest surgery. However, percutaneous options are becoming available with the implementation of the NOGA mapping systems (Gyöngyösi M, et al. *Nat Rev Cardiol.* 8:393-404 (2011)). Moreover, our future studies will explore the potential of vascular delivery CMMPs (e.g. intracoronary, intravenous) with the focus on targeting CMMPs to the injury and promoting extravasation."

Reviewer 3

We thank the reviewer for her/his constructive comments which have allowed us to perform the new experiments to improve the study.

1. **The authors state that "CMMPs carried the same secreted proteins and membranes as genuine cardiac stem cells did". However, the data presented look at only a limited set of surface markers (CD105, CD90, CD45, CD34, ckit) and growth factors (VEGF, IGF1, HGF). The same is true of the statement "MPs contained CSC secretome but not the membrane of CSCs" but only 3 growth factors were assessed. Therefore, these conclusions are not substantiated.**

Re: We have toned down the conclusions regarding how CMMPs resemble CSCs in secretome and membrane. Instead of calling "the same", we have replaced them with "a similar" in the revised manuscript.

2. **Related to (1), the authors have investigated only the percent positive CMMPs relative to CSCs. There are no data presented on fluorescence intensity, which would be an indication of density of markers and how this compares between the CMMPs and CSCs.**

Re: This is an excellent suggestion. We have included new data (Suppl. Fig. 3) to compare the fluorescence intensities of two major CSC markers (CD105 and CD90) between CMMPs and CSCs.

3. **in Fig. 1j-l, release profiles are given for VRGF, IGF1 and HGF. This is cited as % release. It would be better to provide details of quantity per particle and compare MP, CMMP and CSCs - currently, there are no data on CSCs, which makes it difficult to evaluate how these particles compare with 'real' cells**

Re: We agree with the reviewer. We have included new data (Suppl. Fig. 4) to compare the actual released proteins from CMMPs and CSCs.

4. **I am confused by the claim that "CMMP... will not elicit immune reactions since they are not real cells". This is factually incorrect since the CMMPs are coated with proteins, sugars and lipids, all of which can readily elicit an immune response.**

Re: We agree with the reviewer. In the revised paper, we have toned down our statement regarding the immune response to CMMP injections.

5) The purpose of Fig 2 is confusing. This only shows that CMMPs interact more effectively with NRCMs than do MPs. The aim of the study seems to be to show that CMMPs are at least as good if not better than CSCs, so why are CSCs not included in the experiments?

Re: The reviewer's point is well-taken. We have added the CSC control in Fig 2.

6) The statement "In vivo degradation of CMMPs were evident as only negligible amount of particles remained in the heart at Day 28 (Supplementray Fig. 3)." Not necessarily correct. The authors do not appear to have checked any other organ systems. It is most likely the CMMPs have been washed away to end up in the spleen, thymus, liver etc. The authors should check these organs to identify distribution and cross compare with animals treated with CSCs

Re: We have added biodistribution data in the submitted revision (Suppl. Fig. 6). Using ex vivo fluorescent imaging, we found the “wash away” of CMMPs to the lung and the liver. This is consistent with previous report that the needle injection can cause vessel damage and venous drainage brings the particles to the lungs (Al Kindi A, et al. *Front Biosci.*13:2421-34 (2008).). The off-target expression in the liver may represent the leakage of CMMPs into the LV cavity during injection. Nevertheless, the majority of CMMPs remains in the heart after injection.

7) The analysis of the animal hearts is rather superficial. For example, the reader is told only that there is viable myocardium of scar tissue in Fig. 3. There is no information on potential mechanism of action, e.g. is there remuscularisation to account for the large differences in ventricular wall thickness? If this effect is anti-apoptotic, some insight into the protective nature is needed. Also, if CMMPs are more stable, easier to cryopreserve and transport than CSCs, there need to be convincing data that show this over either a time course of storage or at least no cryo vs cryo.

Re: We agree with the reviewer that mechanistic insights regarding how CMMPs work are important. The new Fig. 4 includes data on the effects of CMMP injection on remuscularisation (Fig. 4a), proliferation (Fig. 4b), and angiogenesis (Figs. 4c & d). Protective effects were also evident: CMMPs are anti-apoptotic (data included in the original Fig. 3d). We did include a no cryo vs cryo comparison in Suppl. Fig. 5 to evaluate the stability of CMMPs undergoing cryopreservation.

8) The purpose of Figure 4 is not clear. This shows in a zebrafish model that CMMPs can escape through the vasculature. However, there is no biological significance to this experiment. Can CMMPs home to the heart or do they become spread throughout the various organ systems in the body? Is this further improved by localised delivery to the vasculature of the heart? These questions need a detailed investigation that would be better saved for a separate manuscript. The authors should focus on solidifying whether CMMPs can improve heart function by direct injection and of they can what is the mechanism of action

Re: We agree with the reviewer regarding the premature nature of the zebrafish data. The editors concurred with the reviewer and actually suggested removing this dataset. To those ends, we have removed the zebrafish data but replaced with a new Fig 4 to focus on the direct injection of mechanisms of CMMPs.

Nevertheless, we have added Discussion regarding the possible delivery routes for CMMPs.

On Page 10 Line 12 (Discussion Part), we added: “CMMPs will most likely be delivered intramyocardially via direct muscle injection. Such injection normally requires open-chest surgery.

However, percutaneous options are becoming available with the implementation of the NOGA mapping systems (Gyöngyösi M, et al. *Nat Rev Cardiol.* **8**:393-404 (2011)). Moreover, our future studies will explore the potential of vascular delivery CMMPs (e.g. intracoronary, intravenous) with the focus on targeting CMMPs to the injury and promoting extravasation.”

Minor:

- The text and fig 3a say LVEF are measured after 4 weeks but fig 3k says 3 weeks

Re: We have corrected this typo. It should be 4 weeks.

- There are typographical errors throughout the manuscript

Re: We really apologize for the typo errors. We have hired a native English speaker to check the language and hopefully it is now acceptable.

Reviewer #1 (Remarks to the Author):

The revision is significantly improved and all critical comments have been addressed. This is an innovative and impactful study that should be of broad interest. Acceptance is recommended of this manuscript.

--

Reviewer #3 (Remarks to the Author):

The authors have made quite substantial changes, which have improved the manuscript. They have reinforced their data on the biodistribution of the particles and as expected a good deal ends up in other organs.

They have also provided data on the impact of the particles on cell behaviour in the heart - here they cite "remuscularisation (Fig. 4a), proliferation (Fig. 4b), and angiogenesis (Figs. 4c & d)". It is this that I struggle with as the data are not tremendously strong - it is difficult to know whether cardiomyocytes (or their progenitors) really are proliferating and leading to remuscularisation, and if so to what degree.

However, at this stage, I support publication of the manuscript with some discussion of this caveat.

Comments by the reviewers:

Reviewer 1

Reviewer #1 (Remarks to the Author):

The revision is significantly improved and all critical comments have been addressed. This is an innovative and impactful study that should be of broad interest. Acceptance is recommended of this manuscript.

Re: We are grateful to the referee for her/his appreciation of this work.

Reviewer #3 (Remarks to the Author):

The authors have made quite substantial changes, which have improved the manuscript. They have reinforced their data on the biodistribution of the particles and as expected a good deal ends up in other organs.

They have also provided data on the impact of the particles on cell behaviour in the heart - here they cite "remuscularisation (Fig. 4a), proliferation (Fig. 4b), and angiogenesis (Figs. 4c & d)". It is this that I struggle with as the data are not tremendously strong - it is difficult to know whether cardiomyocytes (or their progenitors) really are proliferating and leading to remuscularisation, and if so to what degree.

However, at this stage, I support publication of the manuscript with some discussion of this caveat.

Re: We appreciate the constructive comments from reviewer which have allowed us to improve the study. We agree with the reviewer that we should discuss this caveat in the paper.

In Discussion section, we added "One caveat of our study is that with the existing assay it is difficult to conclude whether cardiomyocytes (or their progenitors) really are proliferating and leading to remuscularisation after CMMP injection."